# Determination of SLES in Personal Care Products by Colloid Titration with Light Reflection Measurements

**DOI:** 10.3390/molecules26092716

**Published:** 2021-05-05

**Authors:** Dorota Ziółkowska, Iryna Syrotynska, Alexander Shyichuk, Jan Lamkiewicz

**Affiliations:** 1Faculty of Chemical Technology and Engineering, UTP University of Science and Technology, Seminaryjna 3, 85-326 Bydgoszcz, Poland; szyjczuk@utp.edu.pl (A.S.); potas@utp.edu.pl (J.L.); 2Biochemistry Department, Ivano-Frankivsk National Medical University, Halyts’ka 2, 76000 Ivano-Frankivs’k, Ukraine; irenka.933@gmail.com

**Keywords:** quantitation of surfactants, turbidity, polyDADMAC, PDADMAC, PDDA, cationic polymer

## Abstract

The method of colloid titration with poly(diallyldimethylammonium) chloride has been improved to detect the endpoint with an off-vessel light reflectance sensor. The digital color sensor used measures light reflectance by means of light guides, with no immersion into the reaction solution. In such a method, the optical signal is free of disturbances caused by sticky flocs in the solution. The improved automatic titration set was applied for the determination of sodium laureth sulfate (SLES) in industrial batches and commercial personal care products. The sample color and opacity do not disturb the SLES quantification. When the SLES content lies in the range from 5% to 9%, the optimal sample weight is from 6 g to 3 g.

## 1. Introduction

Sodium laureth sulfate (SLES) is a common primary surfactant in skin and hair care products [1,2,3]. Therefore, the determination of SLES concentration in personal care products is a routine task in industrial labs.

The simplest method for the determination of anionic surfactant is the methylene blue active substances (MBAS) procedure. The MBAS method is based on the formation of strong associates of anionic surfactants with methylene blue dye, followed by extraction with chloroform or dichloromethane and measurement by vis-photometry [4,5]. In order to avoid the use of toxic solvents, a titration procedure using methylene blue dye was developed [6]. Advanced instrumental techniques, such as high-performance liquid chromatography (HPLC) [5,7], gas chromatography–mass spectrometry (GC–MS) [8] and others [7,9] are also employed for surfactants analysis. These expensive methods are less suitable for use in small laboratories. The most common method is a titration with a cationic surfactant. Typically, the endpoint is detected by means of surfactant-selective electrodes. In the titration procedure, oppositely charged surfactants form insoluble ion associates. The inversion of surfactant type in solution in the endpoint results in a drop in potentiometric signal, which is used in the control loop of an automatic titrator. The magnitude and steepness of the potential drop depend on the electrode characteristics, molecular structure of the surfactant, titration rate, solution pH and ionic strength [10,11,12]. The main disadvantage of ion-selective electrodes is a signal disturbance when an insoluble surfactant adheres to the sensor surface. Another method of endpoint detection is turbidity measurement [13]. However, immersed optical probes can also suffer from the adherence of sticky surfactant aggregates. 

The very suitable titrant for determining anionic surfactants proved to be poly(diallyldimethylammonium) chloride [14]. This cationic polymer (usually referred to as polyDADMAC, PDADMAC or PDDA) is well known to form strong associations with anionic surfactants [14,15]. The main role it plays is electrostatic attraction due to a high cationic charge density on polyDADMAC macromolecules. Alongside this, hydrophobic interaction provides a secondary association resulting in the formation of various globular architectures [13,16,17,18]. In general, hydrophobic interaction and secondary aggregation increase with the increase of the hydrophobic chain length in the surfactant molecule [19]. The self-association of polyDADMAC with anionic molecules is used to tune the adsorption of active cosmetic components on hair and skin surfaces [3,15,20,21]. 

It has been proved that mixtures of polyDADMAC and sodium lauryl sulfate (SDS) with nearly equimolar charge ratios form highly turbid colloidal dispersions, which settle for 1–3 days as a fluffy sediment [22]. This phenomenon was a basis for the determination of SDS by polyDADMAC titration, performed by using turbidity measurements in transmission mode [14]. Both the immersion optical probe and off-vessel optical transmission sensor proved to be fully applicable for endpoint detection. The new procedure turned out to be less labor-intensive compared to the standard MBAS procedure, and less expensive than the chromatographic methods. On the other hand, it requires calibration, which must be repeated each time the standard titrant solution is changed. The method was tested using reagent-grade SDS and resulted in good linearity over a broad determination range [14]. However, the analysis of commercial products revealed a shortcoming of the method. Namely, some personal care products have quite high initial turbidity, which makes it difficult to measure light transmission. Therefore, in the present work, the turbidity of the reaction suspension was determined by measuring the intensity of reflected light. In such a way, the intrinsic turbidity of the sample becomes only a background for the light reflection measurements. The required information is provided by flashes of light caused by surfactant-polymer aggregates flying in the reaction mixture. The improved turbidimetric technique was applied to the determination of SLES samples with a different number of ethoxy groups.

## 2. Results and Discussion

### 2.1. Colloid Titration Method

The basis of the method used is the phenomenon of flocculation of anionic surfactant by a strong cationic polymer. When mixing the surfactant solution with the polymer solution at nearly equivalent amounts, a bulk precipitate arises and the light reflectance varies substantially. To illustrate the colloid titration method, three industrial SLES samples were used. The SLES samples have average numbers of oxyethylene groups in the range from 1 to 3 (Table 1). This degree of ethoxylation is typical of the SLES used in commercial shampoos [2]. The number of ethoxy groups in the SLES molecule proved to have no effect on the surfactant aggregation with polyDADMAC [13]. This advantage is due to the high charge density and stiffness of the polyDADMAC macromolecule.

The example solutions were prepared by the dissolution of SLES in distilled water with no additives. Initially, the SLES solutions were fully clear (Figure 1—column 0). They turned cloudy immediately after polyDADMAC was added (Figure 1—column 20). The solution turbidity arises due to the formation of fine particles of sparingly soluble surfactant–polymer associates [15,22,23]. At low polyDADMAC doses, surfactant anions remain in excess and stabilize the formed polymer-surfactant associates. The mechanism is that surfactant anions are adsorbed on the associates due to hydrophobic attraction. The added charge provides repulsion between micelles, due to the fact that like charges repel one other. As a result, the micelles remain suspended (Figure 1—column 20). An increase in the polyDADMAC amount resulted in an increase in suspension concentration and a corresponding increase in turbidity (Figure 1—columns 30, 40). It can be concluded that further addition of polyDADMAC results in a decrease in free surfactant anions and a decrease in adsorbed charge. The polymer–surfactant associates lose excess anionic charge and, hence, became less stable. When the molar amount of polymer cations to surfactant anions approaches unity, the charges become balanced. Without the charge, hydrophobic micelles lose stability and coagulate in the form of irregular flocs (clearly visible in Figure 1—columns 50 and 60). An excessive amount of polyDADMAC did not affect the size and stability of the formed flocs. The phenomenon of mutual coagulation of anionic surfactants and cationic polymers is described in the literature [13,14,19]. 

To confirm the formation of strong ion associates, the formed precipitates were collected, dried and studied with X-ray diffractometry. The diffractogram of the SLES1 sample contains multiple peaks of moderate height (Figure 2a). Diffractograms of the SLES2 and SLES3 samples contain a strong peak at 19.4 degrees, several small peaks and a broad halo in the range from 17 to 28 degrees (Figure 2b,c). In turn, the structure of polyDADMAC is more ordered. XRD pattern of polyDADMAC contains two strong peaks at 2 theta angles equal to 31.35 and 45.14 degrees (Figure 2). These peaks are in full agreement with the literature data [24,25]. The diffractogram of polyDADMAC (Figure 2) also reveals bumps at 16.15, 21.4 and 27.1 degrees, which are difficult to ascribe. 

The reaction between polyDADMAC and SLES results in substantial changes in diffraction patterns—all the characteristic peaks of polyDADMAC and SLES have disappeared. Instead, two new narrow peaks are formed at 38.1 and 44 degrees (Figure 2), indicating the formation of a new ordered structure. Alongside this, a broad halo pattern in the range from 15 to 28 degrees indicates that the new structure has a rather semi-crystalline character. Thus, the registered substantial changes in molecular ordering confirm the ionic reaction of polyDADMAC with SLES. The formed polymer–surfactant associates reveal rather short-range ordering.

The points of equimolar coagulation of SLES anions by polyDADMAC polycations were determined by using photometric signal changes. The exemplary photometric titration plots are presented in Figure 3. The range of photometric sensor signal values depends on the quantity of surfactant–polymer flocs, as well as the geometry of the measuring system and the operating parameters of the light source. In order to eliminate the influence of apparatus factors, the initial signal graphs were transformed into normalized photometric graphs and more sophisticated derivative graphs. A normalized signal was obtained by dividing the difference of the instantaneous and average sensor signals by standard deviation. The shapes of the original and normalized titration curves are identical. Initially, an increase in the titrant volume resulted in an increase in the photometric signal. The obvious cause is the increase in the reaction mixture turbidity. The greater number of colloidal particles, the more light is reflected by the suspension. In the vicinity of the equimolar ratio of polyDADMAC to SLES, the sensor signal decreases rapidly (Figure 3). The cause is the coagulation of the colloidal suspension, resulting in a decrease in the number of suspended particles and a corresponding decrease in the amount of reflected light. Alongside this, the average size of particles is increased. Further addition of polyDADMAC results in flocculation (Figure 1—columns 50 and 60). The flying flocs cause multiple signal oscillations. Single giant agglomerates can cause large spikes on the titration graphs (Figure 3). 

The obtained titration graphs are quite different from the ones registered in light transmission mode [14]. At the initial stage of titration, light reflection increases (Figure 3) while light transmission decreases [14]. At the endpoint, however, signal oscillations are clearly detected in both the reflection and transmission modes. The titration endpoint may be determined in various ways as shown in Figure 3. According to method A, the endpoint corresponds to the first appearance of signal oscillations (○). It can be read directly from the original titration graph as well as from the transformed or derivative one. Method A does not provide reliable detection at low analyte concentrations because signal oscillations are too low. Method B uses the absolute value of the derivative of the original signal. The endpoint is the point where the derivative signal deviates from a straight line (●). In fact, the derivative signal enables the use of both methods A and B to detect the endpoint. 

A series of titration curves of standard samples show a clear relationship between the concentration of the sample and the onset of the signal oscillation (Figure 4). Both the methods for endpoint determination produce rectilinear dependencies between the SLES content and titrant amount (Figure 5). However, method B results in higher values of the coefficient of determination, R^2^. As a matter of fact, method B provides the value R^2^ > 0.9985 (Figure 5). In other words, the deviation of the signal derivative provides more precise endpoint determination than the appearance of optical signal oscillations. The same conclusion was reached after testing the repeatability of the methods (Table 2). Thus, method B is more precise.

One more observation applies to the slope values of the calibration lines (Figure 5). Actually, the numerical values of slopes (in mg/mmol) should be equal to molecular weights of the SLES samples (in g/mol). In the case of SLES1, the slope value (Figure 5, method B) and the molecular weight value (Table 1) are in good agreement (348.7 and 340, respectively). In the case of SLES2, the numerical values are slightly different (Mw and slope are equal to 400 and 384, respectively). In the case of SLES3, the values of Mw and slope differ markedly (391 and 435, respectively). Probably, these differences arose as a result of accidental changes in the chemical composition of industrial batches.

Figure 6 shows the effect of interfering substances, which are typically present in personal care products. Without the additives, the recovery of the method equals about 101–102%. However, the addition of electrolytes or non-ionic surfactants results in a gradual decrease in recovery values. Adding sodium chloride in a concentration of 40 mM decreased determination recovery to 93% (Figure 6a). The adverse effect of sodium chloride on the binding of SDS anions to polyDADMAC cations is well known [17,26,27]. Its mechanism is the electrostatic shielding of both the organic ions by inorganic counterions. Sodium chloride in concentrations 100–200 mM leads to a substantial decrease in SDS association with polyDADMAC (at SDS concentration 0.1–1 mM [26] and 4–6 mM [27]). At a sodium chloride concentration equal to 200 mM and SDS concentration equal to 0.5–8 mM, no stable colloidal dispersion was observed [17]. Citric acid has a milder interfering effect than sodium chloride (Figure 6b). The recovery value was above 95% until the concentration of citric acid reached 100 mM. In turn, the interfering effect of non-ionic surfactants is stronger than that of sodium chloride (Figure 6c,d). The amount of non-ionic surfactants needed to decrease the recovery to 95% ranges from 0.75 to 1, relative to the amount of SLES. It has been proved that non-ionic surfactants markedly affect the turbidity of polyDADMAC-SDS colloid solutions with an equimolar ratio of the components as well as with SDS excess [18,28,29]. The onset of a turbidity decrease is reported to be at a fourfold concentration of n-dodecyl-β-D-maltoside surfactant, as compared to SDS [28]. The underlying mechanism is the formation of ternary mixed associates. The combined action of 300 mM sodium chloride with 25–40 mM of n-dodecyl-β-D-maltoside converts polyDADMAC-SDS precipitate into a stable colloidal solution [29].

### 2.2. The Determination of SLES in Commercial Products

The improved method of colloid titration with light reflectance measurements was applied to the determination of SLES in commercial detergent and personal care products. Typically, the SLES content in shampoo ranges from 5% to 15% m/m [2,23]. Considering the range of determination of the method in question, the sample masses were established to be 1 g, 3 g, 6 g and 10 g (Table 3).

The first example (sample S1) is laundry liquid with a fairly simple formula. The product had the appearance of a cloudy blue liquid (Appendix A). According to the manufacturer’s information (Table 4), this product contains SLES (with <2.5 of ethylene oxide groups per molecule), sodium dodecylbenzene sulfonate, and non-ionic surfactants as well as preservatives (benzisothiazolinone and sodium pyrithione). The diluted sample remained slightly turbid (Appendix A). Titration with polyDADMAC resulted in gradually increased turbidity followed by flocculation. The obtained photometric lines (Figure 7a) are very similar to those of standard SLES solutions (Figure 4). Sharp fluctuations on the derivative plots allow the precise determination of the endpoint (Figure 7b). The fluctuations are less distinct at lower sample mass (1 g) while increased sample masses provide quite distinct endpoints. The obtained mean value of SLES content equal to 5.5% (Table 3) agrees well with the manufacturer’s information (5–15%). It can be concluded that the blue color and the initial turbidity of the sample do not interfere with the determination procedure. However, too low a sample mass can lead to the overestimation of SLES content.

The second example is hair shampoo S2 (Table 4). The product was a transparent greenish liquid (Appendix A). Apart from SLES, the shampoo contains cocamide DEA and cocamidopropyl betaine (secondary surfactants) as well as decyl glucoside, laurdimonium hydroxypropyl (hydrolyzed wheat protein), laurdimonium hydroxypropyl (hydrolyzed wheat starch) and laureth-7 citrate. Titration with polyDADMAC resulted in typical changes of turbidity: a gradual increase in photometric signal was followed by a steep fall and then oscillations (Figure 8a). The obtained photometric graphs are consistent with the changes in the appearance of the sample during titration (Appendix A). The graphs are also similar to those of standard SLES solutions (Figure 4). Although the derivative plots possess distinct endpoints at all the sample masses applied (Figure 8b), it seems that the result obtained for the 1 g sample mass was overestimated (Table 3).

The hair shampoo S3 was a white, milky liquid (Appendix A). Its composition included eighteen ingredients (Table 4). The main ingredients are: SLES, cocamide DEA, sodium chloride, glycol distearate, and laureth-4. The cationic polymer polyquaternium-10 is listed among the minor components. Cationic polymers are used in hair care products to improve the adsorption of active components on the hair surface [20,21]. Probably, polyquaternium is merely the component that forms the milky turbidity. The low-soluble anti-dandruff agent, zinc pyrithione, can also contribute to turbidity. Under the titration of the samples with polyDADMAC, the light reflectance signal is increased and then slightly decreased, and next the fluctuations arose (Figure 9a). The changes in turbidity prior to precipitation were not observed visually (Appendix A). The shape of the photometric titration curves differs from the previous ones in that no clear maximum is observed. Alongside this, derivative graphs contained sharp fluctuations, allowing the precise determination of the endpoint (Figure 9b). 

The only sample mass of 1 g results in a photometric graph with a slight fluctuation. Nevertheless, the obtained SLES contents are similar for all sample masses (Table 3). One can conclude that the initial turbidity of the product does not interfere with SLES determination. However, the presence of the cationic polymer polyquaternium-10 may result in the underestimation of the SLES content.

The hair shampoo S4 was a white, milky pearlescent liquid (Appendix A). The diluted samples also remained cloudy, which made it difficult to observe changes in turbidity during the analysis. The composition of the shampoo S4 contains as many as twenty-nine ingredients (Table 4). The main ones are SLES, cocamidopropyl betaine, dimethiconol, sodium chloride, and cetearyl alcohol. The cationic polymer polyquaternium-10 is again present. Light reflection photometry under the titration procedure was difficult because of foaming (Appendix A). The obtained photometric titration graphs (Figure 10a) are quite different as compared to the previously described products: the initial part of the curve is not as smooth as in the previous examples, no sharp drop in signal is observed and the signal fluctuations in the final part of the graph are rather moderate. Despite this, the derivative graphs allow the determination of the endpoints at sample masses of 1 g, 3 g and 6 g (Figure 10b). Probably, the result obtained for the 1 g sample is overestimated (Table 3). In turn, the SLES content was not determined for the 10 g sample mass because of a lack of large fluctuations on the titration graph. Therefore, when the SLES content is above 8%, the sample mass should be in the range of from 3 g to 6 g.

The hair shampoo S5 was also a white milky liquid (Appendix A). The main ingredients are SLES, cocamidopropyl betaine, sodium chloride, PEG-3 distearate. The shampoo S5 contains two cationic polymers: guar hydroxypropyltrimonium chloride and polyquaternium-10 (Table 4). The diluted samples remained turbid, and foam occurred when the dilute sample was stirred (Appendix A). These features made it difficult to observe changes in turbidity during the analysis. The photometric titration graphs contain no sharp drop in signal (Figure 11a). However, signal fluctuations are quite clearly visible in the final part of the graph. The derivative graphs provide reliable end-points (Figure 11b). The SLES percentages are determined for sample masses of 3 g and 6 g (Table 3). The titration endpoint at a sample mass of 10 g exceeds the range of determination. Thus, the optimal sample mass ranged from 3 g to 6 g.

The baby shampoo S6 was a transparent yellow liquid (Appendix A) having rather a simple formula (Table 4). Among ten ingredients, the main surface-active ones are SLES, laureth3 and cocamidopropyl betaine. Titration of the shampoo S6 resulted in continuously increased turbidity, followed by sudden flocculation (Appendix A). The registered photometric graphs (Figure 12a) are very similar to those of the standard SLES solutions (Figure 4). The endpoints are clearly visible on the modified titration graphs (Figure 12b). The calculated SLES percentages at sample masses of 1 g, 3 g and 6 g are similar (Table 3). Due to the high content of SLES, the titration endpoint at a sample mass of 10 g does not fall within the determination range.

The baby shampoo S7 was a transparent yellowish liquid (Appendix A). The shampoo formula contains no SLES (Table 4). The main ingredient is hydrogenated starch hydrolysate acting as a skin moisture-retaining agent. The surface-active agents are skin-friendly ones: amphoteric cocamidopropyl hydroxysultaine and cocamidopropyl betaine as well as non-ionic lauryl glucoside, coco-glucoside and glyceryl oleate. The anionic surfactant hydrogenated palm glycerides citrate is listed as a minor component. During titration of the shampoo S7 with polyDADMAC, slight turbidity arose but no precipitation was observed (Appendix A). Probably, it is due to the minor component Hydrogenated Palm Glycerides Citrate that forms colloidal associates with polyDADMAC. The formed colloidal suspension remained stable because of the low content of the anionic surfactant and the far larger content of amphoteric and non-ionic surfactants. As a result, the registered photometric graphs are quite flat (Figure 13a). Some signal changes are observed with the addition of very small amounts of polyDADMAC (Figure 13a,b), which are below the lower limit of the range of determination. The obtained SLES content is almost zero (Table 3), which is in line with the manufacturer’s declaration (Table 4).

The intimate hygiene gel (S8) was a transparent blue liquid (Appendix A) with a long formula. Among twenty-three ingredients (Table 4), the primary surfactant is SLES. The auxiliary surfactants are the amphoteric lauramidopropyl betaine, and the non-ionic ones, lauryl glucoside, coco-glucoside, glyceryl oleate and PEG-75 lanolin. The anionic surfactant hydrogenated palm glycerides citrate and the amphoteric surfactant lecithin are listed among the minor ingredients. The quaternary ammonium salt undecylenamidopropyltrimonium methosulfate serves as an antimicrobial agent. Ascorbyl palmitate, with a slightly anionic character, serves as an antioxidant. During titration of the samples S8 with polyDADMAC, turbidity was first increasing and then decreased (Figure 14a), following the changes in the appearance of the sample (Appendix A). The first part of the photometric graph corresponds to the formation and thickening of the suspension. The second part corresponds to the dilution of the suspension. The suspension remained stable and no sediment was observed (Appendix A). As a result, no noticeable fluctuations were registered on the titration graph. The shape of the titration graphs is quite different from those of the standard SLES solutions (Figure 4). However, the distinct peaks on the modified titration graphs (Figure 14b) allow the endpoint determination. The SLES contents determined at sample masses of 3 g, 6 g and 10 g are close to each other (Table 3). Similarly to other products with a low SLES content, the result for the smallest sample mass is rather overestimated. 

The shower oil (S9) was a transparent dark-yellow liquid (Appendix A). The only surfactants are laureth-4 and SLES (MIPA-laureth sulfate) (Table 4). The third listed ingredient is *Ricinus communis* seed oil. After tenfold dilution, the product S9 turned cloudy due to an emulsion being formed (Appendix A). Titration of the sample S9 with polyDADMAC resulted in a gradual increase of turbidity, followed by a steep fall (Figure 15a). The surfactant–polymer aggregates were rather poorly distinguishable (Appendix A). The registered titration graphs (Figure 15a) are similar in shape to those of the reference SLES samples (Figure 4), although without pronounced oscillations after the endpoint. The conclusion is that a significant oil content disturbs the formation of large flocs. For that reason, the fluctuations of the photometric signal are rather small (Figure 15a). Nevertheless, the modified titration curves (Figure 15b) make it possible to precisely determine the endpoint. The calculated SLES percentages are the highest among all the examined products (Table 3). Therefore, the results of a titration at sample masses of 6 g and 10 g are beyond the range of determination.

## 3. Materials and Methods

### 3.1. Materials

A 20% aqueous solution of poly(diallyldimethylammonium) chloride with Mw 100,000–200,000 and reagent-grade sodium dodecyl sulfate (≥99%) were obtained from Sigma Aldrich. The industrial-grade SLES samples were sulfated ethoxylated alcohols C12–C14 from PCC Exol SA (Table 1). Industrial-grade non-ionic surfactants Rokanol L5P5 and Rokanol LP700 were alkoxylated propoxylated linear alkyl alcohols from PCC Exol SA. Reagent grade sodium chloride and citric acid were obtained from POCh (Poland). Commercial personal care products were purchased from a local store. The ingredients of the personal care products are listed in Table 4. The products differed in terms of color and turbidity (Appendix A).

### 3.2. Methods

Titration experiments were carried out by means of an automatic burette Titronic 500 (SI Analytics, Germany) controlled by a PC. The turbidity of the reaction mixture was measured using an off-vessel optical sensor, CROMLAVIEW^®^ CR100 (ASTECH GmbH, Germany) working in reflective mode. A cold white LED was used as the source and the green component of the reflected beam was used as the analytical signal. Both data acquisition and burette control were performed using ChemiON software, written by Jan Lamkiewicz [30]. The analyte solution with a volume of 50 mL was titrated against standardized 50 mM polyDADMAC solution, with 0.02–1 mL increment and 10 s time steps under continuous stirring. The titrant solution was standardized against reagent-grade SDS. No pH adjustment was made, due to the fact that polyDADMAC retains a high dissociation degree in a broad pH range [13]. The ion associates formed at equimolar ratio were collected, dried and examined with X-ray diffractometry. The XRD patterns were recorded using a SEIFERT diffractometer with a CuKα source and a nickel filter.

## 4. Conclusions

A new method of anionic surfactant determination, based on the association of an analyte with an oppositely charged polymer, was applied for the determination of SLES in personal care products. The use of the external light detector in the reflectance mode allowed for the examination of both colored and turbid samples. To find optimal conditions for the analysis, multiple titration curves at various sample masses were recorded and compared. The most appropriate form of titration curves proved to be a derivative graph. It was found that the color of the product does not interfere with the results of SLES determination. In turn, the opacity of the analyzed product or high content of oil components can result in a change of titration curve shape. Fortunately, the changed shape of the titration curve does not make the analysis impossible. The SLES determination results may depend on the sample mass taken for the analysis. In the case that the SLES content is less than 10% w/w, the optimal sample mass is about 3–6 g. When the sample mass is less than 3 g, the SLES content may be underestimated. In the case that the SLES content is above 10%, the optimal sample weight is about 2 g.

## Figures and Tables

**Figure 1 molecules-26-02716-f001:**
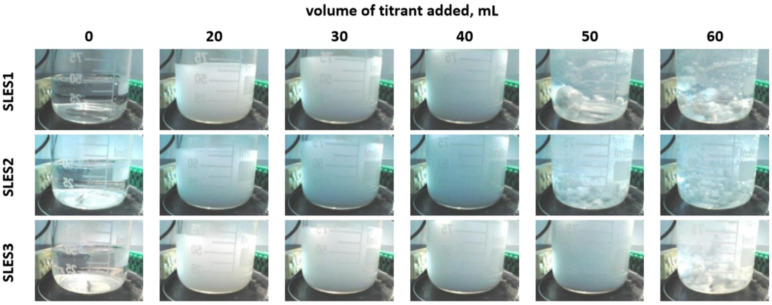
Images captured during the titration of the standard solutions of SLES with a concentration of 40 mM.

**Figure 2 molecules-26-02716-f002:**
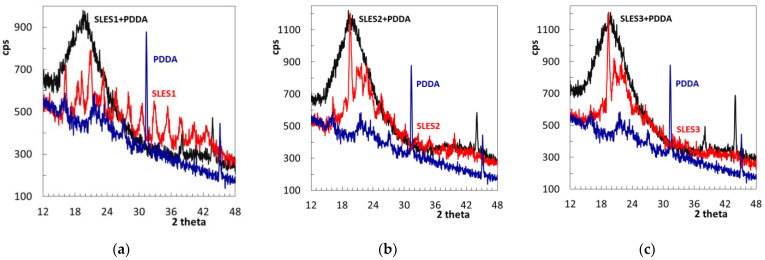
XRD patterns of the titrant and SLES1 (**a**), SLES2 (**b**) and SLES3 (**c**) samples as well as the corresponding reaction products.

**Figure 3 molecules-26-02716-f003:**
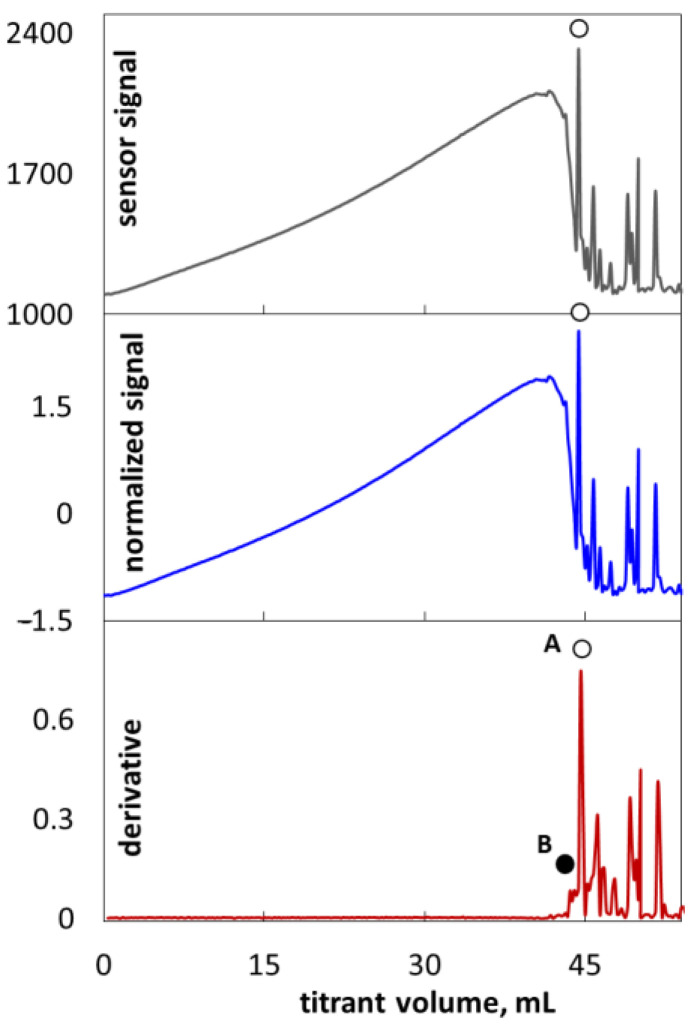
The examples of initial, normalized and derivative photometric plots. The signs ● and ○ indicate two methods of the end-point determination.

**Figure 4 molecules-26-02716-f004:**
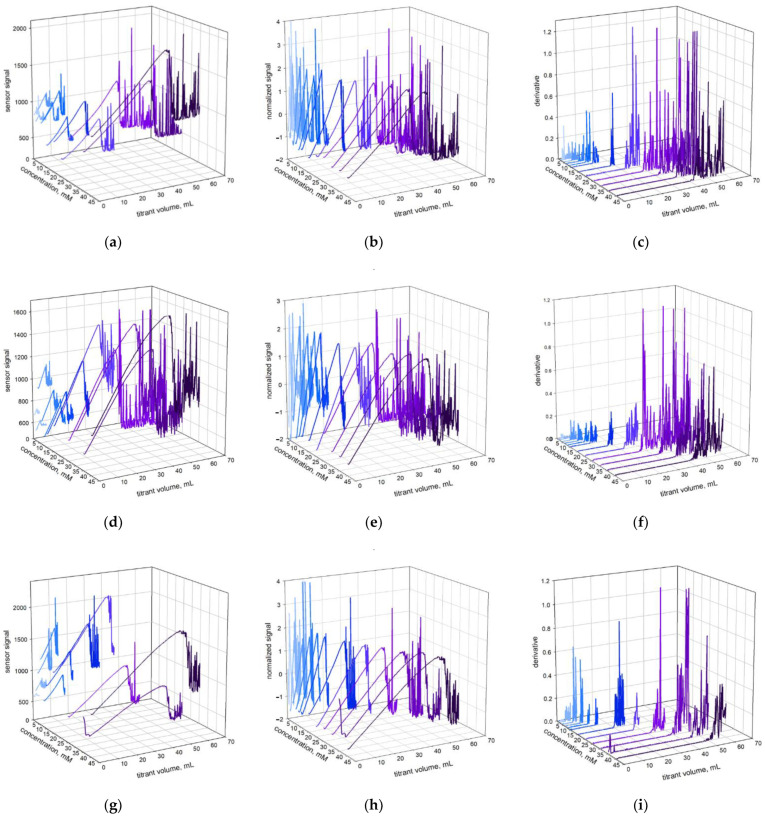
Photometric signal changes during titration of SLES1 (**a**–**c**), SLES2 (**d**–**f**) and SLES3 (**g**–**i**) at indicated concentrations, shown as initial (**a**,**d**,**g**) and normalized (**b**,**e**,**h**) plots as well as derivative ones (**c**,**f**,**i**).

**Figure 5 molecules-26-02716-f005:**
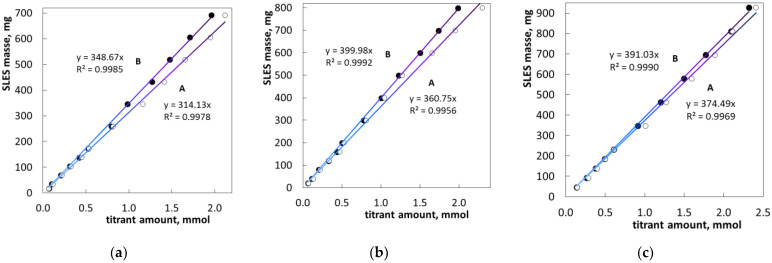
Mass of SLES1 (**a**), SLES2 (**b**) and SLES3 (**c**) vs. the molar amount of standardized PDDA solution. The signs ● and ○ correspond to the two methods of endpoint determination explained in Figure 3.

**Figure 6 molecules-26-02716-f006:**
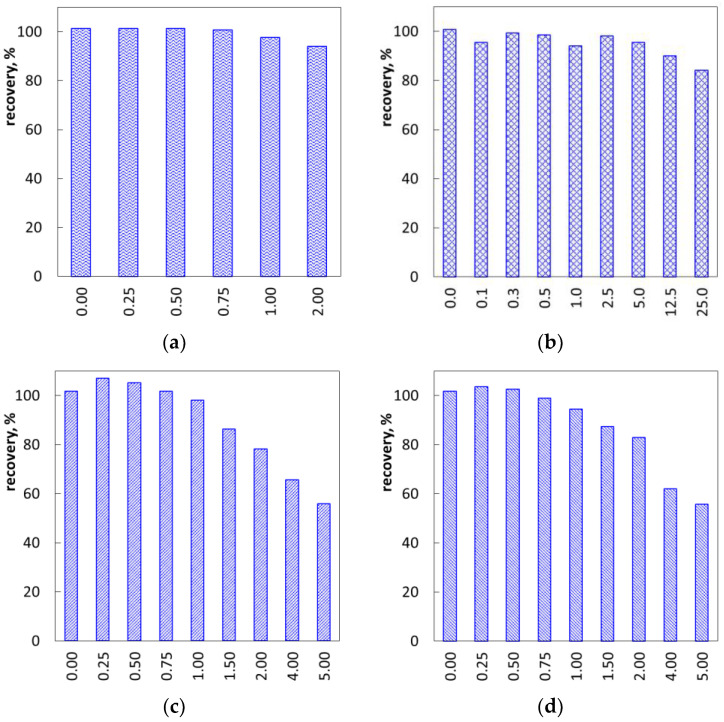
Recovery of SLES determination depending on sodium chloride (**a**), citric acid (**b**), Rokanol L5P5 (alcohol) (**c**), and Rokanol LP700 (ether) (**d**); the amount of SLES concentration, 40 mM. The numbers are molar ratios of the interference substances to SLES.

**Figure 7 molecules-26-02716-f007:**
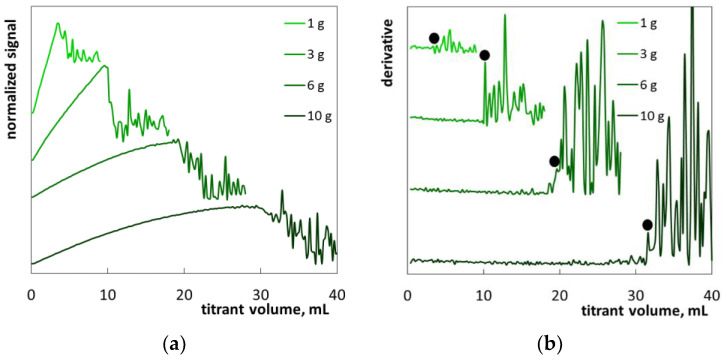
Photometric graphs (**a**) and corresponding derivative graphs (**b**), registered under the SLES determination in the laundry liquid (S1) at indicated sample masses.

**Figure 8 molecules-26-02716-f008:**
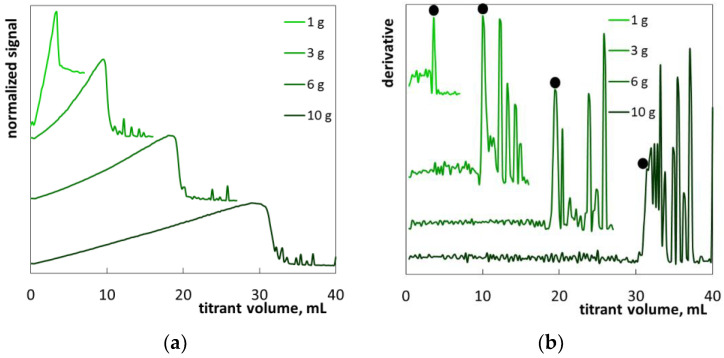
Photometric graphs (**a**) and corresponding derivative graphs (**b**), registered under the SLES determination in the hair shampoo S2 at indicated sample masses.

**Figure 9 molecules-26-02716-f009:**
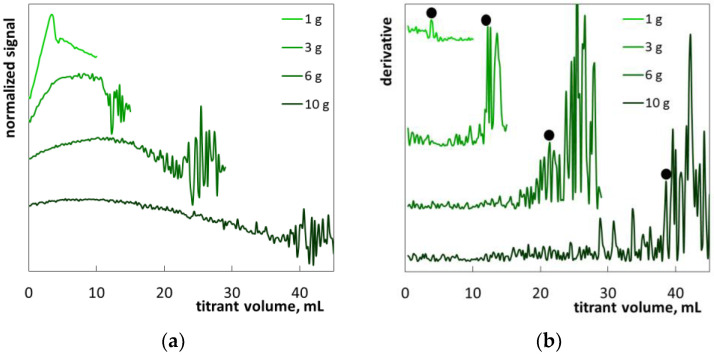
Photometric graphs (**a**) and corresponding derivative graphs (**b**) registered under the SLES determination in the hair shampoo S3 at indicated sample masses.

**Figure 10 molecules-26-02716-f010:**
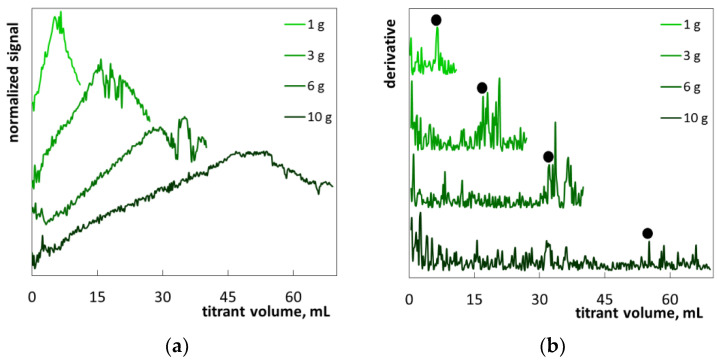
Photometric graphs (**a**) and corresponding derivative graphs (**b**) registered under the SLES determination in the hair shampoo S4 at indicated sample masses.

**Figure 11 molecules-26-02716-f011:**
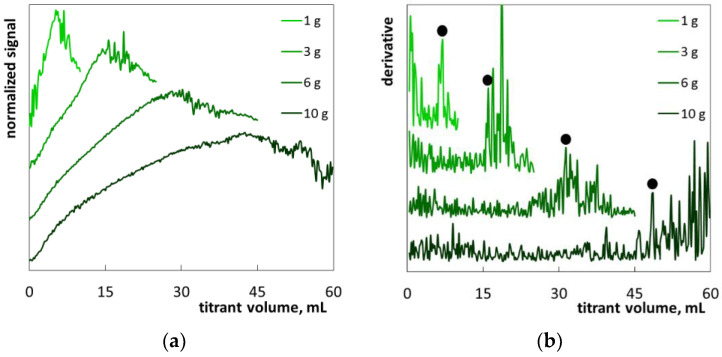
Photometric graphs (**a**) and corresponding derivative graphs (**b**) registered under the SLES determination in the hair shampoo S5 at indicated sample masses.

**Figure 12 molecules-26-02716-f012:**
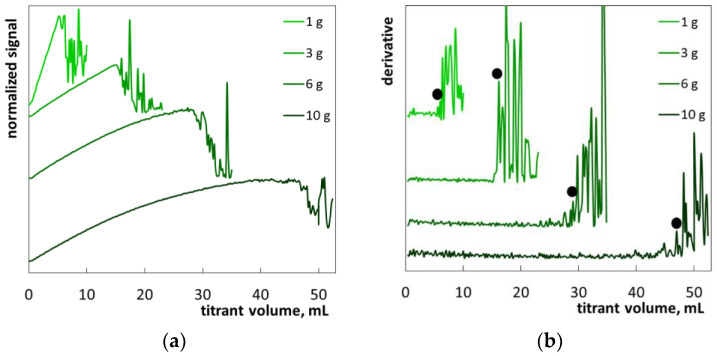
Photometric graphs (**a**) and corresponding derivative graphs (**b**) registered under the SLES determination in the baby shampoo S6 at indicated sample masses.

**Figure 13 molecules-26-02716-f013:**
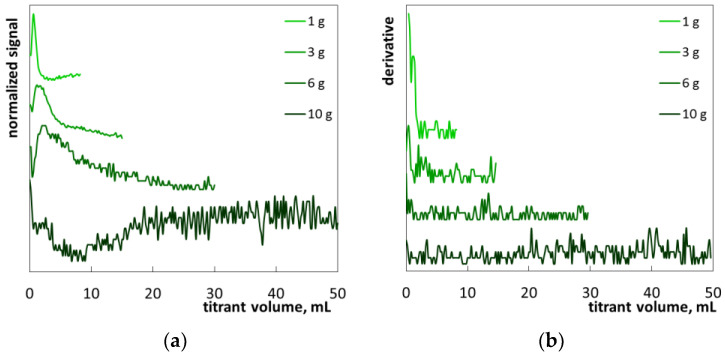
Photometric graphs (**a**) and corresponding derivative graphs (**b**) registered under the SLES determination in the baby shampoo S7 at indicated sample masses.

**Figure 14 molecules-26-02716-f014:**
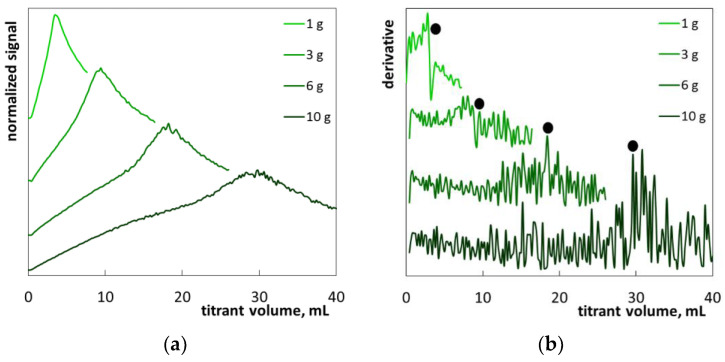
Photometric graphs (**a**) and corresponding derivative graphs (**b**) registered under the SLES determination in the intimate hygiene gel S8 at indicated sample masses.

**Figure 15 molecules-26-02716-f015:**
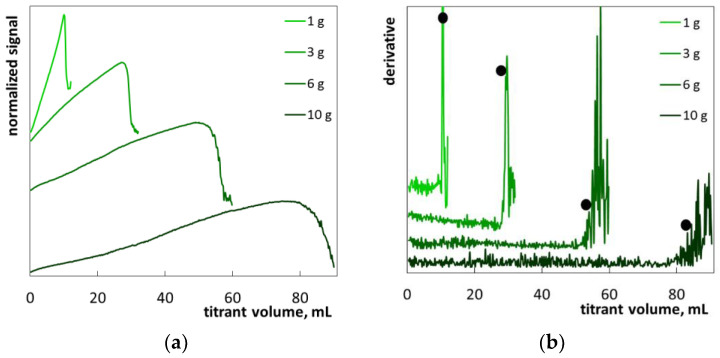
Photometric graphs (**a**) and corresponding derivative graphs (**b**), registered under the SLES determination in the shower oil S9 at indicated sample masses.

**Table 1 molecules-26-02716-t001:** Characteristics of the SLES samples used.

Sample Code	Brand Name	Values from the Manufacturer’s Leaflet			Measured Dry Mass, (% m/m)
		Ethoxy Group Content	Molecular Weight, (g/mol)	Active Substance Content, (% m/m)	
SLES1	SULFOROKAnol L170/1	1–2.5	approx. 340	68–72	71.2
SLES2	SULFOROKAnol L270/1	1–2.5	approx. 384	68–72	72.8
SLES3	SULFOROKAnol L370	>2.5	approx. 435	68–72	74.6

**Table 2 molecules-26-02716-t002:** Relative standard deviation of SLES quantification in 20 mM solution.

Sample	Method	Average ^1^, mM	Variance ^1^	SD ^1^, mM	RSD ^1^, %
SLES1	A	18.73	1.01	1.00	5.36
SLES1	B	20.09	0.03	0.17	0.85
SLES2	A	19.17	0.29	0.53	2.79
SLES2	B	20.24	0.03	0.18	0.88
SLES3	A	20.23	0.14	0.38	1.86
SLES3	B	20.38	0.02	0.12	0.60

^1^ The values were determined from a series of 10 repeated titration experiments.

**Table 3 molecules-26-02716-t003:** Determination of SLES content in commercial products.

Sample	SLES Determined in mmol/g at Different Sample Masses				Average Content, mmol/g	Average Content, % m/m ^1^
	1 g	3 g	6 g	10 g		
S1 laundry liquid	(0.170)	0.152	0.148	0.141	0.147	5.5
S2 hair shampoo	(0.161)	0.149	0.145	0.141	0.145	5.5
S3 hair shampoo	0.170	0.182	0.160	0.173	0.171	6.4
S4 hair shampoo	(0.286)	0.254	0.242	b/r	0.248	9.3
S5 hair shampoo	(0.286)	0.239	0.225	b/r	0.232	8.7
S6 baby shampoo	0.251	0.239	0.216	b/r	0.235	8.9
S7 baby shampoo	~0	~0	~0	~0	0	0
S8 intimate hygiene gel	(0.170)	0.143	0.139	0.132	0.138	5.2
S9 shower oil	0.474	0.442	b/r	b/r	0.458	17.2

^1^ The mass content was calculated by using the value of average molar mass 376.5 g/mol corresponding to SLES molecule with two ethoxy groups. The reason is that SLES with two or one ethoxy groups are typical components in hair care products [13]. b/r—beyond the range of determination ( )—values not included in the mean.

**Table 4 molecules-26-02716-t004:** Ingredients of the personal care products studied.

Sample Code	S1	S2	S3	S4	S5	S6	S7	S8	S9
Type of Product	Laundry Liquid	Hair Shampoo	Hair Shampoo	Hair Shampoo	Hair Shampoo	Baby Shampoo	Baby Shampoo	Intimate Hygiene Gel	Shower Oil
aqua	●	●	●	●	●	●	●	●	●
Sodium Laureth Sulfate	●	●	●	●	●	●		●	
MIPA-Laureth Sulfate									●
Sodium C10-13 Alkyl Benzenosulfonate	●			●					
Laureth-7 Citrate		●							
Hydrogenated Palm Glycerides Citrate							●	●	
Cocamidopropyl Hydroxysultaine							●		
Cocamidopropyl Betaine		●	●	●	●	●	●		
Lauramidopropyl Betaine								●	
Betaine								●	
Cocamide MEA				●					
Cocamide DEA		●	●						
Laureth 2		●							
Laureth 3						●			
Laureth-4			●						●
Ceteareth-18				●					
Trideceth-10				●					
PEG-3 Distearate				●	●				
PEG-40 Hydrogenated Castor Oil					●				
PEG-75 Lanolin								●	
PEG-14M		●							
PEG/PPG-120/10 Trimethylolpropane Trioleate		●							
Lauryl Glucoside							●	●	
Coco-Glucoside							●	●	
Decyl Glucoside		●							
Glyceryl Oleate							●	●	
Glycol Distearate			●	●					
Laurdimonium Hydroxypropyl Hydrolyzed Wheat Protein		●							
Laurdimonium Hydroxypropyl Hydrolyzed Wheat Starch		●							
Hydrolyzed Silk				●	●				
Hydrolyzed Milk Protein					●				
Hydrogenated Starch Hydrolysate							●		
Glycerol			●		●		●		
Propylene Glycol		●						●	●
Panthenol				●		●		●	
Allantoin								●	
Urea		●							
Biotin		●							
Polyquaternium-10			●	●	●				
Guar Hydroxypropyltrimonium Chloride					●				
Lecithin								●	
Sodium Chloride			●	●	●	●	●		
Tetrasodium EDTA		●							
Disodium Phosphate		●							
Citric acid		●	●	●	●	●	●	●	●
Lactic Acid							●	●	
Sodium Hydroxide				●	●				
Aloe Barbadensis Leaf Juice								●	
Gossypium Herbaceum Seed Extract		●							
Prunus Amygdalus Dulcis Seed Extract							●		
Macadamia Ternifolia Seed Oil				●					
Glicine Soja Oil									●
Ricinus Communis Seed Oil									●
Dimethiconol				●					
Lanolin Alcohol					●				
Cetearyl Alcohol				●					
Tocopherol							●	●	
BHT									●
Propyl Gallate									●
Ascorbyl Palmitate								●	
Parfum	●	●	●	●	●	●	●	●	●
Alpha-Isomethyl Ionone									●
Geraniol			●	●	●				●
Hexyl Cinnamal			●	●					
Butylphenyl Methylpropional			●						
Limonene	●		●	●					●
Linalool			●	●	●				●
Citronellol					●				●
Coumarin				●					
Benzyl Alcohol				●	●				●
Benzyl Salicylate				●	●				
Caprylic/Capric Triglyceride		●							
CI 19140						●			
CI 42090								●	
CI 75810		●							
Benzophenone-4								●	
Diethylhexyl Syringylidenemalonate		●							
Zinc Pyrithione			●						
Sodium Pyrithione	●								
Potassium Sorbate				●				●	
Phenoxyethanol				●		●			
Methylparaben						●			
Propylparaben						●			
Methylchloroisothiazolinone		●	●						
Methylisothiazolinone		●	●						
Benzisothiazolinone	●								
DMDM Hydantoin		●				●			
Sodium Benzoate				●	●			●	
Benzoic Acid				●					
p-Anisic Acid							●		

## Data Availability

All data are contained within the article and Appendix A.

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
