# Peer review of "Determination of SLES in Personal Care Products by Colloid Titration with Light Reflection Measurements"

_molecules, 2021, doi:10.3390/molecules26092716_

Round 1
Reviewer 1 Report
The paper is well written and the work was well done as well. However, I suggest adding into the introduction section other well-known methods able to determines SLES. For example, authors can introduce the spectrophotometric method MBAS (Methylene Blue Active Substances), and/or the chromatographic determination by LC/MS-MS. Moreover, authors can compare the method reported in the present work with other ones by reporting the advantages/disadvantage with respect to other methods.
Author Response
Thank you very much for the positive evaluation of our article.
According to the Reviewer's suggestions, the paragraph on the use of MBAS and chromatographic methods as well as a sentence on the advantages and disadvantages of new procedure comparing to other methods have been added to the introduction.
On behalf of the authors,
Dorota ZióÅ‚kowska
Reviewer 2 Report
The manuscript describes an improved method of quantitative analysis of sodium LaurEth sulfate in commercial skin and hair acre products by a titration with cationic surfactant. End point is detected by measuring light reflectance in a non-contact way which reduces disturbances.
The manuscript is well written and provides sufficient details of every aspect of the research. The improved method has a potential to find a way in routine analyses.
Comments:
1. Why low sample mass yields overestimated values of SLES content?
2. Indicate end point on graphs 7-15 as is indicated on the Fig 3.
How are noisy signals treated?
3. What is the wavelength of the source?
I recommend publication of the manuscript in the Molecules journal after addressing abovementioned comments.
Author Response
Thank you very much for the positive evaluation of our article.
Our responses for Reviewer’s questions are as follows:
- Why low sample mass yields overestimated values of SLES content?
At low surfactant amount, a small excessive amount of polyDAMAC is needed in order the cloudy suspension is formed. This is probably why the titrant volume causing significant changes in sensor signal is somewhat overestimated. In general, reading from the lower and higher parts of the standard curve gives less accurate results than reading from the middle part.
- Indicate end point on graphs 7-15 as is indicated on the Fig 3. How are noisy signals treated?
The graphs 7-15 were supplemented with marked end-points.
The signal fluctuations come from the finest particles of the suspension. According to the end-point definition B, it is read at a point where the signal clearly deviates from the mean value of these fluctuations. In some cases, a small difference in noise height and analytical signal height makes it difficult but not impossible to recognize the end-point.
- What is the wavelength of the source?
A cold white LED was used as the source. The applied illuminating intensity was 409 lux (40% of maximum value). The reflected beam was split into RGB components. Component G was used as the analytical signal. A sentence on the source has been added to the introduction.
On behalf of the authors,
Dorota ZióÅ‚kowska